# Uptake of Biotinylated Spermine in Astrocytes: Effect of Cx43 siRNA, HIV-Tat Protein and Polyamine Transport Inhibitor on Polyamine Uptake

**DOI:** 10.3390/biom11081187

**Published:** 2021-08-11

**Authors:** Christian J. Malpica-Nieves, Yomarie Rivera, David E. Rivera-Aponte, Otto Phanstiel, Rüdiger W. Veh, Misty J. Eaton, Serguei N. Skatchkov

**Affiliations:** 1Department of Biochemistry, Central University of the Caribbean, Bayamon, PR 00956, USA; yomarie.rivera@uccaribe.edu (Y.R.); david.rivera@uccaribe.edu (D.E.R.-A.); misty.eaton@uccaribe.edu (M.J.E.); 2Department of Medical Education, University of Central Florida, Orlando, FL 32816, USA; otto.phanstiel@ucf.edu; 3Institut für Zell- und Neurobiologie, Charité, 10117 Berlin, Germany; ruediger.veh@charite.de; 4Department of Physiology, Central University of the Caribbean, Bayamon, PR 00956, USA

**Keywords:** astrocytes, HIV-Tat, polyamines, acetylated polyamines, astroglial polyamine transporter, spermine catabolism, Cx43 gap junctions and hemichannels, diseases

## Abstract

Polyamines (PAs) are polycationic biomolecules containing multiple amino groups. Patients with HIV-associated neurocognitive disorder (HAND) have high concentrations of the polyamine N-acetylated spermine in their brain and cerebral spinal fluid (CSF) and have increased PA release from astrocytes. These effects are due to the exposure to HIV-Tat. In healthy adult brain, PAs are accumulated but not synthesized in astrocytes, suggesting that PAs must enter astrocytes to be N-acetylated and released. Therefore, we tested if Cx43 hemichannels (Cx43-HCs) are pathways for PA flux in control and HIV-Tat-treated astrocytes. We used biotinylated spermine (b-SPM) to examine polyamine uptake. We found that control astrocytes and those treated with siRNA-Cx43 took up b-SPM, similarly suggesting that PA uptake is via a transporter/channel other than Cx43-HCs. Surprisingly, astrocytes pretreated with both HIV-Tat and siRNA-Cx43 showed increased accumulation of b-SPM. Using a novel polyamine transport inhibitor (PTI), trimer 44NMe, we blocked b-SPM uptake, showing that PA uptake is via a PTI-sensitive transport mechanism such as organic cation transporter. Our data suggest that Cx43 HCs are not a major pathway for b-SPM uptake in the condition of normal extracellular calcium concentration but may be involved in the release of PAs to the extracellular space during viral infection.

## 1. Introduction

The human immunodeficiency virus (HIV), identified in 1980, is a lentivirus that attacks the immune system and brain of humans and other primate species [1], also known as simian immunodeficiency virus (SIV). HIV-1 is highly virulent and infective [2] and is responsible for the AIDS pandemic [3]. The HIV-1 genome contains three structural genes: gag, pol, and env; two regulatory genes: tat and rev; three accessory genes: nef, vif, and vpr; and at least seven structural elements. The initial steps of HIV infection are binding of the virions to the CD4 cell receptor and co-receptor on the cell surface. These co-receptors are called CCR5 and CXCR4 [4] and play an important role in the initiation and spread of the HIV infection. R5 viruses are the most common HIV strains isolated from HIV infected central nervous system (CNS) [5], and the brain is the second most frequently infected organ in HIV-infected individuals [6]. Therefore, in addition to affecting the immune system, HIV infection is responsible for causing cognitive, behavioral, and motor dysfunction including a direct pathology in the brain, spinal cord, and peripheral nerves [7].

Since combined highly active antiretroviral therapies (HAART) became the standard treatment for HIV, life expectancy has increased [7]. Nevertheless, around half of individuals infected with HIV-1 still have cognitive impairment [8]. The term HIV-associated neurocognitive disorder (HAND) has been used to describe the spectrum of neurocognitive dysfunction associated with HIV infection. In these patients, HAART only prevents HIV progression to AIDS, but does not protect these individuals from developing HAND. This could be possible due to the neurotoxic actions of HIV proteins or toxic products released from astrocytes due to the presence of viral proteins such as the HIV-1 trans-activator of transcription (HIV-Tat) [9]. The detrimental effects of HAND on survival, quality of life, and everyday functioning make it an important unresolved issue in HIV-positive patients.

HIV-Tat is a protein made by the HIV virus that can promote efficient transcription and replication of the HIV genome [10,11]. Moreover, this protein can cause cell death via the interacting mechanisms at the glutamatergic sites (NMDA receptors) in which Tat causes the release of Zn^2+^ from its binding site, resulting in NMDA receptor activation and increasing its capacity to allow calcium ion flux [12,13]. HIV-Tat can also cause downstream activation of proinflammatory regulatory chemokines such as nuclear factor kappa-light-chain-enhancer of activated B (NF-κB) [14,15]. Exposure to Tat protein somehow causes neuronal dysfunction and promotes cell death in vitro and in vivo [15,16].

A key finding by Merali and co-authors (2014) showed that astrocytes are the principal cells in which HIV-1 Tat protein activates the enzyme spermidine/spermine acetyltransferase (SSAT) that converts the polyamine (PA) spermine (SPM) to N^1^-acetyl-SPM. Taking into account that: (i) glial cells lack the biosynthetic enzymes for PA synthesis [17,18] but accumulate PAs [19,20,21], and (ii) astrocytes are HIV reservoirs [22,23,24], we can assume that astrocytes are key players in HAND. Astrocytes are rich of PAs, while viruses, RNA, and DNA have a unique capability to interact with PAs [25,26]. Taken together, these findings highlight the interactions where HIV-1 Tat protein produces conversion of SPM in astrocytes to N-acetylated-SPM (Ac-SPM), which is a novel biomarker predicting the severity of HAND in HIV-infected individuals [27]. Therefore, we suggest that SPM can be a physiological gliotransmitter [28,29] and Ac-SPM is a pathological gliotransmitter correlating with HAND pathology [27]. The release of PAs was observed in the brain [30] and nerves [31]. Ultimately, PAs are taken up by glia, where they can be the source of release to modulate both the glial and neuronal channels [32,33]. Since glia outnumber neurons [34] and PAs increase longevity [35,36], glial cells are the focus of the current research.

Since PA uptake, N-acetylation, and release are key aspects of HAND, it is important to understand how PAs enter cells. On the basis of previous publications on PA permeability via large pores such as connexin-43 (Cx43) [37,38,39] and published data on PA transport [40], we identified two strong candidate proteins for PA transport in astrocytes: organic cation transporters (OCTs) and Cx43 hemichannels (HCs). OCTs are transmembrane proteins that belong to the solute carrier family 22 (SLC22A1-3) of polyspecific facilitative transporters. They mediate the uptake, distribution, and efflux of organic cations including PAs across the cell membrane [40,41,42]. Of this family of proteins, OCT3 is expressed in astrocytes [43] and has been shown to transport PAs [40,42]. Glial Cx43 HCs are another type of transmembrane protein formed by six proteins called connexins. Cx43 HCs allow for the transfer of ions and small molecules less than 1–2 kDa, making these another potential pathway for the PA uptake. To understand how changes in the dynamics of PA flux can be related to the development of HAND, one needs to identify the PA uptake pathways and their effect in the astrocytic syncytium. Our preliminary electrophysiological, RNAi, and optical imaging data suggest that Cx43 HCs and gap-junctions and OCTs are potential pathways for PAs. Furthermore, we found that PAs potentiate opening of astrocytic Cx43 HCs and gap junctions, when the calcium concentration is in the micromolar range [37,38,39].

The purpose of the present study was to determine if Cx43 HCs and/or OCTs are major transmembrane proteins involved in PA uptake in astrocytes to replenish the pool of PA for their N-acetylation in patients with HAND. To study the potential pathways for PA uptake in astrocytes expressing HIV-Tat protein, we incubated cultured astrocytes with SPM tagged with biotin (biotinylated SPM, b-SPM (see Section 2 below)) and visualized b-SPM uptake using fluorescent rhodamine-avidin (biotin–avidin binding) with confocal microscopy. Cx43 HCs were knocked down using siRNA, whereas potential PA uptake via transporters was blocked using a novel polyamine transport inhibitor (PTI) that we have characterized in astrocytes [33]. To mimic HAND conditions, we exposed cultured astrocytes for 72 h prior to experimentation to HIV-1 IIIB Tat Recombinant Protein.

Our results indicate that in normal millimolar concentrations of divalent cations (Ca^2+^, Mg^2+^), PAs enter the astrocytes via a PTI-sensitive pathway, but not via Cx43 HCs. Interestingly, HIV-Tat protein increases the b-SPM uptake/accumulation in Cx43 siRNA-treated astrocytes, but not in mock-transfected astrocytes and siRNA Cx43-treated cells without HIV-Tat. This suggests the possibility that Cx43 HCs act as a release pathway for PAs in the presence of HIV-Tat. Taken together, these data suggest that Cx43 is not an uptake pathway for SPM when divalent cations are in the steady state range but may be involved in the release of PAs to the extracellular space, particularly during viral infection.

## 2. Materials and Methods

### 2.1. Animals

Experiments were carried out in accordance with a protocol approved by the Universidad Central del Caribe Institutional Animal Care and Use Committee. A breeding colony was established using C57BL/6N (C57) mice obtained from Charles River Laboratories (Wilmington, MA, USA).

### 2.2. Preparation of Primary Cortical Astrocyte Cultures

Primary cultures of astrocytes were prepared from the neocortex of 1–3-day-old C57 mice, as previously described [33]. Briefly, brains were removed after decapitation, and the meninges were stripped to avoid fibroblast contamination. Mixed glial cultures were plated in Dulbecco’s modified Eagle’s medium (DMEM) containing 25 mM glucose, 2 mM glutamine, 1 mM pyruvate, 10% fetal calf serum, and 100 U/mL penicillin/streptomycin. At 85–90% confluence, cultures were treated with 50 mM leucine methyl ester (LME) in growth medium (pH 7.4) for 60 min to eliminate microglia. Cultures were then allowed to recover for at least 2 days in growth medium prior to experimentation.

### 2.3. SDS-PAGE and Western Blotting Analysis

After 4 weeks in culture, cortical astrocytes were harvested, pelleted, and resuspended in homogenization buffer as previously described [33]. Briefly, Western blotting was performed using anti-Connexin-43 antibody produced in rabbit (1:2000; #C6219, Sigma Aldrich, St. Louis, MO, USA) and 15 µg/mL of protein from each sample. The homogenization buffer (pH 7.5) contained (in mM): 20 Tris–HCl, 150 NaCl, 1.0 EDTA, 1.0 EGTA, 1.0 phenylmethylsulfonyl fluoride (PMSF), 1% Triton X-100, and an additional mixture of protease inhibitors (leupeptin, bestatin, pepstatin, and aprotinin). The membranes were stained with India ink for total protein to quantify small differences in sample loading. Final detection was performed with the enhanced chemiluminescence methodology (SuperSignal^®^ West Dura Extended Duration Substrate; Rockford, IL, USA) as described by the manufacturer, and the signal was quantified using a gel documentation system (ChemiDoc, Bio Rad, Hercules, CA, USA). The image was obtained using the Image Lab software (Version 6.1, Bio-Rad, Hercules, CA, USA).

### 2.4. siRNA Transfection

After 4 weeks in culture, astrocytes were trypsinized and plated 50,000 cells per well in a 24 well plate in which cover glasses were already placed in each well. Immediately after plating, cells were transfected with 20 nM of siRNA targeting Cx43 (Catalog number SI03071726 from GeneGlobe, Qiagen, Germantown, MD, USA) and 1.2 µL of HiPerFECT (Catalog number 30170 from GeneGlobe, Qiagen, Germantown, MD, USA) in 500 µL of cell culture media (DMEM high glucose without FBS and antibiotic, see above for recipe), according to the manufacturer’s instructions. Transfected astrocytes were used for experiments after 72 h. Prior to the experiments, we determined the optimal time and concentration of siRNA necessary to obtain the maximal decrease in Cx43 expression (data not shown).

### 2.5. HIV-Tat Protein

HIV-1 IIIB Tat (HIV-Tat) Recombinant Protein (Catalog number 160018) was obtained from the NIH AIDS Reagent Program (NIH, Germantown, MD, USA). The protein was reconstituted in phosphate-buffered saline (CPBS), which contained in mM: 133 NaCl, 2.7 KCl, 8.2 Na_2_HPO_4_·7H_2_O, and 2.2 NaH_2_PO_4_, pH 7.4. Cells were treated at the time of transfection with 50 ng/mL of HIV-Tat 72 h prior to the experiment.

### 2.6. Polyamine Transport Inhibitor (PTI)

The trimer44NMe is a novel polyamine transport inhibitor (PTI) [33,44,45] that was developed and synthesized in the laboratory of Dr. Otto Phanstiel at the University of Central Florida in Orlando, FL. The PTI has been used in combination with difluoromethylornithine (DFMO) to deplete PAs in cancer cells [44]. The PTI was diluted using CPBS and used at a concentration of 30 µM. This concentration was chosen on the basis of our optimization studies for primary-cultured astrocytes established in prior work [33] using the LIVE/Dead cell assay (Number L3224, ThermoFisher Scientific, Waltham, MA, USA, Cat.).

### 2.7. Biotinylated Spermine Uptake and Visualization

Biotinylated spermine (b-SPM) was synthesized following established methods to conjugate N-hydroxy-succinimide esters of haptens with amino group-containing molecules [46,47]. In short, 17.4 mg of spermine tetrahydrochloride and 27 µL triethanolamine (to deprotonize SPM) were dissolved in 1 mL of dimethylformamide (50 mM SPM, solution A). Furthermore, 6.8 mg of biotin-N-hydroxysuccinimide ester were dissolved in another 1 mL DMFA (20 mM; solution B). For conjugation, 100 µL of solution A was combined with 150 µL of DMFA and supplemented with 250 µL of solution B. After 2 h at room temperature, the reaction was stopped by the addition of 4.5 mL of pure ethanol, resulting in a 1 mM final concentration of biotinylated spermine.

For the uptake experiments, we removed the media from the transfected cells and washed them with saline solution at 37 °C (in mM, NaCl 140; HEPES 10; KCl 3; MgCl_2_ 2; CaCl_2_ 2.5; Glucose 10). We added 500 µL of the saline solution with 80 µM of b-SPM and incubated the mixture for 1 h at 37 °C with gentle shaking. Experimental groups receiving HIV-Tat were incubated with HIV-Tat protein at the same time as the transfection (72 h prior to the experiment). Experimental groups receiving PTI were treated one hour prior to b-SPM treatment and also included with b-SPM for the one-hour incubation period. After the incubation period ended, the solution was removed, and the cells were rinsed 1 time with 500 µL of cold 4% paraformaldehyde (PFA) in phosphate-buffered saline solution (PBS in mM: NaCl 137; KCL 2.7; Na_2_HPO_4_ 10; KH_2_PO_4_ 1.8). The PFA was exchanged with 1 mL of fresh cold PFA, and cells were incubated at 4 °C for 1 h. Fixation is very important to maintain the fine structure of the tissue, but even more to chemically link the PAs to adjacent proteins. The cells were fixed without a previous washing step, otherwise the PAs would be lost during washing steps. After incubation, the cells were washed 3 times with PBS or with 0.3% Triton X (triton) in PBS at room temperature for 5 min. Cells were then incubated in 500 µL/well of Rhodamine Avidin D (rho-avidin; A-2002, Vector Laboratories, Burlingame, CA, USA) solution (1/200 dilution in PBS) at room temp for 2 h, after which, the cells were washed with PBS for 10 min, 20 min, 30 min, and 1 h at room temperature. These washing steps were critical to decrease non-specific fluorescence due to residual Rhodamine Avidin D that was not attached to the b-SPM. Cell nuclei were stained using DAPI (1/5000 in PBS) for 10 min and subsequently washed 3 times with PBS for 5 min to remove residual DAPI. The cover glasses were carefully removed from the 24 well plate and adhered to microscope slide using Vectashield mounting media (H-1000-10, Vector Laboratories, Burlingame, CA, USA); then, they were dried for 24 h prior to their examination using confocal microscopy.

### 2.8. Confocal Microscopy

The slides were imaged using an Inverted Fluoview-1000 Laser Scanning Confocal Microscope (LSCM) (Olympus, Tokyo, Japan). Six images per slide were obtained by LSCM using Rhodamine Red-X (580 nm) and DAPI (455 nm) lasers and a 60× objective. Laser exposure intensity was the same for all images in a given experiment. Initially, we set the laser exposure intensity using positive control cells and then we switched to a negative control slide to reduce laser exposure to the minimum. Third, we switched back to the positive control slide to see the b-SPM signal and make any last and minor adjustments to the laser intensity before proceeding to capture images from all slides without changing any parameters. Images were analyzed using ImageJ (Version 1.52u, NIH, Bethesda, MD, USA) and Microsoft Excel (version 16.35 for Mac, Microsoft, Redmond, WA, USA). We used the following formula Equation (1) to calculate the cell fluorescence in all the images, taking into account differences in the numbers of cells per image [48].
(1)$$\mathrm{Cell}\;\mathrm{Fluorescence}=\frac{\left[\mathrm{integrated}\;\mathrm{density}\;\mathrm{of}\;\mathrm{image}-\left(\mathrm{area}\;\mathrm{of}\;\mathrm{image}\times \mathrm{fluoresence}\;\mathrm{of}\;\mathrm{background}\;\mathrm{reading}\right)\right]}{\mathrm{number}\;\mathrm{of}\;\mathrm{cells}\;\mathrm{in}\;\mathrm{image}}$$

### 2.9. Statistics

All statistical analyses were performed using GraphPad Prism Version 8.4.3 (471) (San Diego, CA, USA). Data from Western blots were analyzed using an unpaired Student’s *t*-test (Figure 1). Data from the b-SPM experiments were analyzed using a two-way ANOVA, Tukey’s multiple comparison, and Sidak’s multiple comparison tests (Figures 2–4). A one-way ANOVA, Tukey’s multiple comparison, and Dunnett’s multiple comparison tests were used to analyze data in Figure 5. A value of *p* < 0.05 was considered significant.

## 3. Results

### 3.1. Cx43 Downregulation Validation via SDS Page and Western Blot

To assess the efficacy of the siRNA treatment in each experiment, we used Western blot to measure the expression of Cx43 in mock-transfected control astrocytes (HiPerFect with no siRNA) versus those transfected with siRNA targeting Cx43. The samples for Western blot were run in parallel with the astrocytes used for b-SPM uptake. As shown in Figure 1, there was an 87% downregulation of Cx43 in siRNA-treated samples as compared with mock-transfected controls (*n* = 3/group). Although we only show this one example of Cx43 knock-down, Western blot was used to assess knock-down in all siRNA experiments, and it was equally effective in all (data not shown).

### 3.2. Contribution of Cx43 Hemichannels to Biotinylated Spermine Uptake into Astrocytes

In the first series of experiments, we determined the contribution of Cx43 hemichannels to b-SPM uptake into cultured astrocytes. Visualization of 80 µM b-SPM uptake depends upon the high affinity between avidin and biotin [49,50], with the SPM being tagged with biotin and the fluorescent label covalently linked to avidin. After processing the experiment, we measured cell fluorescence for each astrocyte in six images taken from each cover slip. There were multiple negative controls in the experimental design to determine levels of background fluorescence. The first negative control excluded triton in a prewash step. Triton opens the cell membrane to allow penetration of rho-avidin into the cell. By processing the samples without triton, we were able to determine the levels of cell fluorescence that could be due to b-SPM or rho-avidin sticking to the extracellular surface of the astrocyte rather than uptake of b-SPM into the cell. A second series of negative controls was to measure cell fluorescence in the presence of the triton wash step, but in the absence of b-SPM. These negative controls assessed the levels of fluorescence that were seen due to rho-avidin entering the cell and not being completely washed away even though it was not bound to b-SPM. As summarized in Figure 2, the overall levels of fluorescence obtained in the absence of triton was consistent between treatments and relatively low. This correlates with general cell fluorescence and any rho-avidin or b-SPM sticking to the extracellular surface of the cells. There was a notable but relatively small increase in cell fluorescence when triton was used in the absence of b-SPM, which provided a measure of the rho-avidin that enters the cell but is not bound to biotin.

For the experimental groups, we measured cell fluorescence as a semi-quantitative measure of b-SPM uptake into control astrocytes (DMEM), mock-transfected astrocytes (DMEM + 1.2 µL of HiPerFECT), and those transfected with siRNA targeting Cx43 (DMEM + 20 nM of siRNA targeting Cx43 + 1.2 µL of HiPerFECT). We found significant uptake of b-SPM into astrocytes of all three experimental triton groups when compared with the no triton control and the no b-SPM groups. The level of uptake was similar for all three treatments including astrocytes in which Cx43 was knocked down, which suggests that Cx43 is not a major pathway for SPM uptake under these conditions (no statistical difference was found among these groups).

### 3.3. Effect of HIV-Tat on the Uptake/Accumulation of b-SPM

In the next series of experiments, the effect of HIV-Tat protein on uptake/accumulation of b-SPM in astrocytes was examined. We noted that prior work in cancer cells by Mani et al. revealed that HIV-Tat was a potent inhibitor of PA uptake [51], but in cultured astrocytes, HIV-Tat (50 ng/mL) did not inhibit PA uptake. Thus, HIV-Tat was used to simulate HAND conditions and test the effect of this viral protein on the uptake of PAs by astrocytes (Figure 3a–c). We used a similar experimental design, including the negative controls described above. As we found in the previous experiment, b-SPM was taken up and accumulated in control, mock-transfected, and Cx43 siRNA-treated astrocytes. Treatment of astrocytes with HIV-Tat for 3 days prior to the experiment did not affect b-SPM uptake in control and mock-transfected astrocytes (as compared with their corresponding controls). Surprisingly, there was a significant increase in cell fluorescence in astrocytes treated with HIV-Tat and siRNA targeting Cx43. These data suggest that in the presence of the HIV-Tat protein, there is no difference in the b-SPM accumulation (uptake minus potential release from the cell) between the mock samples. However, when the expression of the Cx43 HC protein was silenced in the membrane of the astrocytes, there was a significant increase in cell fluorescence, indicating enhanced b-SPM uptake/accumulation within astrocytes (Figure 3d).

### 3.4. Contribution of Polyamine Transport to Biotinylated Spermine Uptake into Astrocytes

On the basis of the findings of experiments 1 and 2 described above, we found that Cx43 hemichannels did not appear to be a major pathway for PA uptake into astrocytes when the extracellular solution contained normal levels of divalent cations. We have previously shown that trimer44NMe (a novel polyamine transport inhibitor, PTI) can inhibit astrocyte cell proliferation in response to exogenously applied spermidine when de novo synthesis of PAs is blocked [33]. We here assessed the ability of the trimer44NMe PTI (30 µM) to block uptake of b-SPM into astrocytes (Figure 4). As indicated by cell fluorescence, b-SPM was taken up into astrocytes when comparing triton- vs. no triton-treated astrocyte controls. This effect was substantially decreased in astrocytes treated with the PTI (30 µM) 1 h prior to and during the 1 h b-SPM incubation period. These data suggest that PAs are taken up into control astrocytes via a PTI-sensitive uptake pathway.

### 3.5. Effect of PTI on Biotinylated Spermine (b-SPM) Uptake in siRNA Cx43-Treated Astrocytes with and without HIV-Tat Protein

The next experiment (Figure 5) was designed to examine the interaction between HIV-Tat treatment and potential pathways for PA flux including Cx43 hemichannels and PTI-sensitive transport. We used the negative control of triton with no b-SPM since it measured background fluorescence due to rho-avidin both inside and outside the cell. Astrocytes were transfected with siRNA targeting Cx43 or mock transfected 72 h prior to the uptake experiment. At the time of transfection, the appropriate groups were treated with HIV-Tat (50 ng/mL). PTI (30 µM) was added to the samples 1 h prior to and during the 1 h incubation with b-SPM. After the astrocytes and PAs were fixed with PFA, all groups were treated with triton. Consistent with the results of our previous experiments, cell fluorescence for mock-transfected astrocytes that were not given b-SPM (negative control) was low, and fluorescence in mock-transfected astrocytes given b-SPM was elevated, indicating astrocytes take up b-SPM. The response was similar for astrocytes in which Cx43 was knocked down. In contrast, uptake was substantially reduced in astrocytes treated with PTI, suggesting that a PTI-sensitive transporter is responsible for the majority of uptake. Consistent with Figure 3, b-SPM uptake occurred in HIV-Tat-treated astrocytes, and the fluorescence accumulated in the cell was comparable to that seen with mock-transfected astrocytes (Figure 5). Fluorescence intensity was substantially increased in cells that were treated with both Cx43 siRNA + HIV-Tat, and this effect was blocked by the addition of PTI. Taken together, these data suggest that uptake of b-SPM by a PTI-sensitive pathway is necessary to see the additive increase in cell fluorescence observed with both Cx43 siRNA and HIV-Tat treatment.

## 4. Discussion

Although PA levels in brain have been associated with the severity of aging, dementia, neurological syndromes, and HIV-associated neurocognitive disorders (HAND), little is known about PA flux in the brain [52,53]. Indeed, alterations in PA homeostasis have been linked to HAND, stroke, epilepsy, Alzheimer’s disease, Huntington’s disease, and Alexander disease, as well as psychiatric disorders. Astrocytes in the adult brain are the predominant stores for PAs [19], but these cells lack the biosynthetic enzymes needed to synthesize these compounds [17,18,54]. Therefore, PAs must be taken up and accumulated in astrocytes via a yet unidentified pathway. The purpose of the present study was to determine if two major transmembrane proteins localized in the astrocytic membrane (Cx43 hemichannels and/or transporters such as OCTs) serve as the uptake pathway for PAs into astrocytes and how these pathways may play a role in PA flux in HIV-Tat treated astrocytes.

We found that astrocytes import exogenously applied b-SPM in control conditions, as seen by an increase in cell fluorescence upon incubation with rho-avidin. This uptake was blocked by PTI, but not by knockdown of Cx43 hemichannels. The Trimer44NMe PTI is a specific blocker for PA transport, which is a competitive inhibitor of PA uptake [45,55]. We have previously shown that this PTI blocks the uptake of exogenously applied spermidine into astrocytes [33] and the results of the present study indicate that b-SPM uptake into astrocytes is via a PTI-sensitive transport mechanism. A likely candidate to mediate this uptake is the SLC22A organic cation transporter 3 (OCT3). OCTs are sodium-independent transporters that translocate a variety of substances that include monovalent, divalent, and polyvalent cations such as L-histamine and dopamine, as well as the PAs putrescine (PUT), spermidine (SPD), and SPM [40,42,56]. In the brain, extremely low levels of OCT2 mRNA were found, while OCT1 and OCT3 mRNA levels are higher [57]. In astrocytes, immunohistochemical localization has been confirmed only for OCT3 [58]. Biochemical characterization of PA uptake in cerebellar astrocytes has shown that uptake is dependent upon: (i) the membrane potential, but (ii) is not mediated by co-transport with sodium [59]. These characteristics are consistent with OCT3 transporters [60,61,62].

Furthermore, our results suggest that Cx43 HCs are not the main pathway for b-SPM uptake in primary-cultured cortical astrocytes. In the present experiment, we applied exogenous b-SPM to astrocyte cultures in the presence of physiologically high concentrations of divalent cations (extracellular 2.5 mM Ca^2+^ and 2.0 mM Mg^2+^). However, it is well established that Cx43 HCs are predominantly closed when there are physiological levels of these extracellular divalent cations [63,64]. Keeping HCs in a predominantly closed state should protect cell integrity since these are large pores in the membrane that allow the non-selective transfer of ions and small molecules less than 1–2 kDa. The probability of opening Cx43 HCs increases when divalent cation concentrations are reduced [64] or by the presence of mM concentrations of SPM [65]. Future experiments can address whether or not Cx43 HCs become a more pivotal pathway for PA uptake during synaptic activity when neurons are firing rapidly and there are local decreases of these cations, particularly Ca^2+^ [66].

It was previously shown that HIV-Tat protein increases the activity of the PA catabolic enzyme spermidine/spermine N-1 acetyl transferase (SSAT) [58] and increases the concentrations of N-acetylated-PAs (Ac-PAs) in the brain and cerebral spinal fluid of HIV-infected individuals [27]. In cultured human astrocytes, it was shown that HIV-Tat (i) increased activation of SSAT and enhanced N-acetylation of PAs, (ii) promoted PA flux in astrocytes, and (iii) increased release of Ac-PAs without affecting PA levels within astrocytes [27].

In the current study, astrocytes were treated with HIV-Tat protein for 72 h prior to incubation with b-SPM. We found that astrocytes took up b-SPM, but there was no difference between the uptake/accumulation of b-SPM between control astrocytes and astrocytes treated with HIV-Tat. This finding is consistent with the study of Merali et al. (2014), which demonstrated that PA levels in HIV-Tat-treated astrocytes were stable. Surprisingly, when we treated astrocytes with HIV-Tat protein and knocked-down Cx43 in these cells, there was a significant increase in cell fluorescence when compared to HIV-Tat alone, indicating that uptake/accumulation of b-SPM was increased. Furthermore, PTI completely blocked b-SPM uptake/accumulation even in HIV-Tat-treated astrocytes in which Cx43 was knocked down.

In the present study, we examined the steady state levels of PAs at 1 h after uptake. The data obtained do not provide information about b-SPM that could have been taken up into astrocytes and subsequently released, which would have been consistent with the increased PA flux described by Merali et al. (2014). Furthermore, we measured cell fluorescence which is due to the attachment of rho-avidin to the biotin molecule that was tagged to SPM. If the b-SPM molecule is transformed to b-Ac-SPM due to activation of spermine/spermidine acetyl transferase, one would expect to still see an increase in cell fluorescence due to an interaction between rho-avidin and the biotin which is now tagged to Ac-SPM.

Taking into consideration the findings of Merali (2014) and the data obtained in our experiments, we propose the following model (Figure 6). In control astrocytes and in HIV-Tat-treated astrocytes, b-SPM is taken up by a PTI-sensitive uptake mechanism (perhaps via OCT3). In HIV-Tat-treated astrocytes, SSAT is activated and SPM within the cell including b-SPM is converted to Ac-SPM (and perhaps b-Ac-SPM) and released from the cells. The overall levels of PAs within the cells are stable, suggesting that additional b-SPM is taken up into the cell to maintain a stable pool. In the situation where astrocytes are treated with HIV-Tat and Cx43 HCs are also knocked down using siRNA, the b-SPM accumulation (as measured by increased cell fluorescence) is more notable, suggesting that Cx43 HCs are a release pathway for PAs particularly in HIV-Tat-treated astrocytes. This increase in cell fluorescence may be due to the presence of b-SPM and/or “b-Ac-SPM” (via conversion of b-SPM to b-Ac-SPM due to SSAT being highly activated in HIV-Tat-treated astrocytes) and their inability to be released due to knock down of Cx43 HC expression.

The actions of the PAs, SPM, and SPD on neuronal receptors and channels has been well studied [20,21,29,32,37,38,39,40]. Less is known about the actions of Ac-PAs and how increased release of Ac-PAs from astrocytes in HIV-infected individuals could contribute to HAND. Our study examining PA uptake and release in HIV-Tat-treated astrocytes may lead to innovative pharmacological approaches to treat HAND. Blocking uptake of PAs will decrease the pool of PAs in astrocytes to convert to Ac-PAs, whereas blocking release of Ac-PAs could prevent their actions on neuronal receptors and channels. There are currently many molecules, including drugs in clinical use or under investigation, that have been reported to impact the function of Cx43 [67,68,69] or the organic cation transporter OCT3 [70,71]. These drugs may be repurposed to alter the uptake of PAs into astrocytes and release of PAs and Ac-PAs from astrocytes with the potential to prevent/treat HAND.

## 5. Conclusions

In conclusion, our data indicate that PAs enter astrocytes via a PTI-sensitive pathway, and Cx43 is not the main pathway for PA uptake in astrocytes. Import of PAs can be blocked by PTI with and without the presence of HIV-Tat protein. This suggests that b-SPM uptake may be via organic cation transporters or another major transmembrane protein that has not been identified yet [72]. Interestingly, HIV-Tat induced Cx43 expression only in human but not in rodent astrocytes [73], which is consistent with our data (not shown). Furthermore, treatment with HIV-Tat protein increases the b-SPM uptake/accumulation when Cx43 function was blocked (siRNA treated astrocytes). There is also the possibility that PAs such as SPD and modified SPM (BEN-SPM, b-SPM) may activate antizyme-1 and SSAT [74], but if b-SPM does so, this will not affect (i) either b-SPM fluorescence accumulation or (ii) the effect of siRNA targeting Cx43. Finally, the OCT-3 transporter may be a major pathway for b-SPM uptake [70] since the SLC22A-3 is expressed in astrocytes [75] and is nonselective for many drugs, toxins, and endogenous compounds [40,41,42,43,70,71]. Therefore, our data suggest the possibility that Cx43 HCs are a release pathway for PAs in HIV-Tat-transformed astrocytes since b-SPM accumulation is greater in the absence of Cx43 expression. These results are consistent with the data described by Merali et al. [27], in which PA flux was increased and increased concentrations of Ac-PAs were observed in the CSF of HAND patients.

## Figures and Tables

**Figure 1 biomolecules-11-01187-f001:**
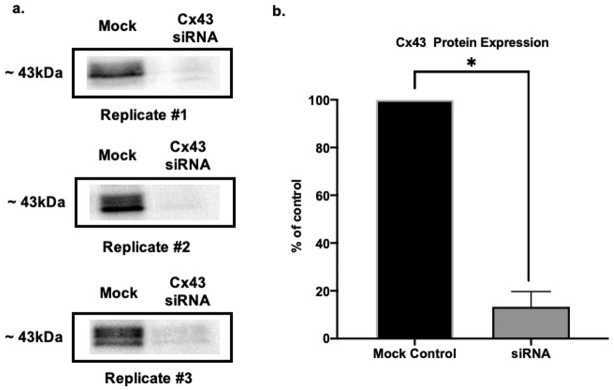
Validation of knockdown of Cx43 expression in astrocytes by siRNA. By using Western blot, levels of Cx43 protein were determined in mock-transfected (transfection reagent with no siRNA) and Cx43 siRNA-transfected astrocytes. Although the data are not shown for each experiment, this validation was performed for each biological replicate of the b-SPM experiments and Cx43 was consistently knocked down. (**a**) Representative Western blot membranes showing knockdown of Cx43 protein expression in primary cultured cortical astrocytes 72 h after transfection with Cx43 siRNA. Cx43 was detected as two bands (phosphorylated and un-phosphorylated) at approximately 43 kDa, and both were used for the calculations shown in the summary. The representative blots show samples from the three biological replicates for the experiment in Figure 2. (**b**) Summary of Cx43 protein expression in mock- and siRNA-transfected astrocytes. Data are expressed relative to mock, which was taken as 100%. The asterisk indicates a significant difference from mock (*t*-test; *p* < 0.05; *n* = 3 for both groups). India ink was used for total protein to discern small differences in protein loading (stained membrane is not shown).

**Figure 2 biomolecules-11-01187-f002:**
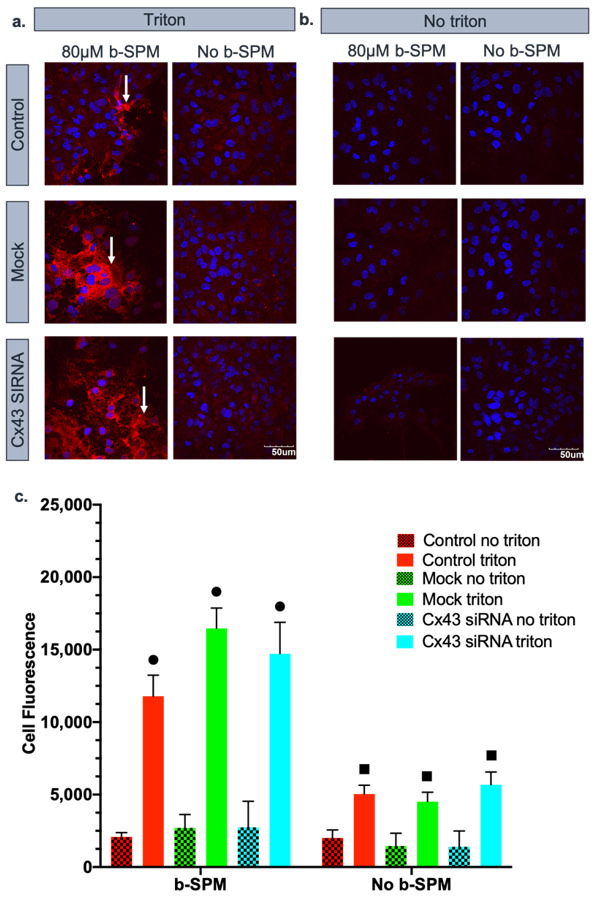
Biotinylated spermine (b-SPM) is taken up into astrocytes. Rho-avidin was used to visualize b-SPM (red), and DAPI was used to visualize the nuclei (blue). The presence of b-SPM presence is marked with the white arrows. (**a**,**b**) Qualitative representation of the data. Control, mock (only treated with HiPerFect), and Cx43 siRNA-transfected astrocytes were incubated in either 80 µM of b-SPM and no b-SPM (PBS) in the presence or absence of triton. No triton and no b-SPM samples were used as negative controls. Calibration bar is the same for all images (50 µm). (**a**) When the membrane was permeabilized with triton, b-SPM was detected in the cytoplasm of the astrocytes in control, mock, and siRNA Cx43 (see white arrows). (**b**) When the membrane was not permeabilized with triton, only minimal fluorescence was seen, indicating that b-SPM was not adhered to the extracellular side of the astrocytic membrane. (**c**) Quantitative representation of the data (*n* = 3/group). Black circles represent (•) statistical difference (*p* < 0.05) between the triton and no triton samples. Black squares (■) represent statistical difference (*p* < 0.05) between b-SPM and no b-SPM samples. There was no statistical difference between mock triton and Cx43 siRNA triton.

**Figure 3 biomolecules-11-01187-f003:**
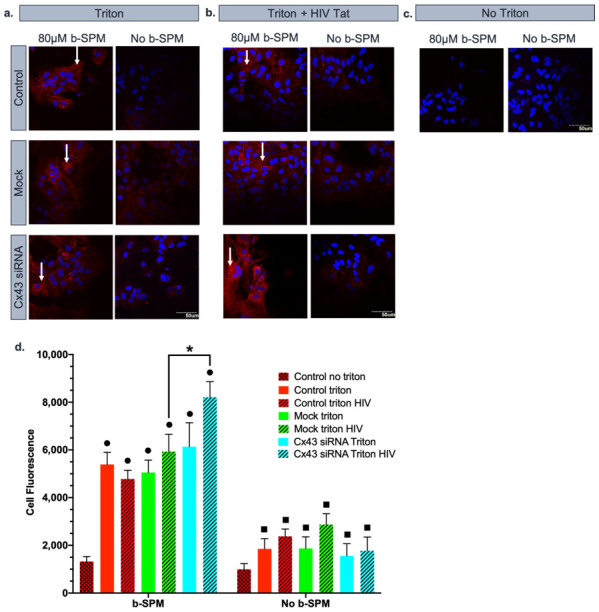
HIV-Tat further increased b-SPM uptake/accumulation only when Cx43 HC were knocked down. Rho-avidin was used to visualize b-SPM (red), and DAPI was used to visualize the nuclei (blue). The presence of b-SPM is marked with the white arrows. (**a**,**b**) Qualitative representation of the data (*n* = 3/group). Control, mock (only treated with HiPerFect), and siRNA specific to Cx43 (+/−HIV-Tat) astrocytes were incubated in either 80 µM of b-SPM and no b-SPM with (PBS). (**c**) No triton and no b-SPM samples were used as negative controls. Calibration bar is the same for all images (50 µm). (**d**) Quantitative representation of the data (*n* = 3/group). Asterisk (*) indicates a significant difference (*p* < 0.05) mock triton HIV and Cx43 siRNA triton HIV. Black squares (■) represent samples statistically significant (*p* < 0.05) with b-SPM samples. Black circles (•) are the samples that are statistically significant with control no triton.

**Figure 4 biomolecules-11-01187-f004:**
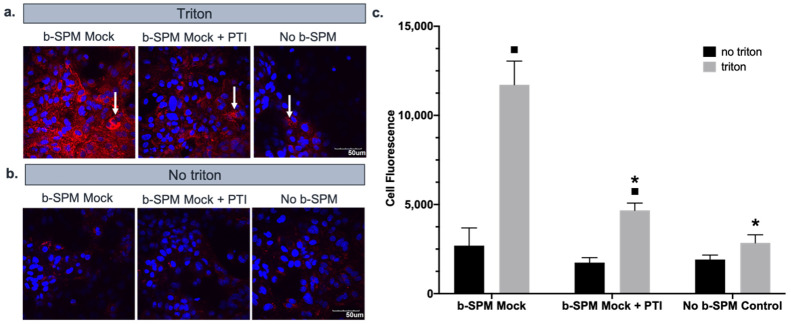
PTI significantly reduced the b-SPM uptake. Rho-avidin was used to visualize b-SPM (red), and DAPI was used to visualize the nuclei (blue). The presence of b-SPM presence is marked with the white arrows. (**a**,**b**) Qualitative representation of the data. Mock (only treated with HiPerFect) and mock + PTI astrocytes were exposed to 80 µM of b-SPM and no b-SPM (PBS). No triton and no b-SPM samples were used as negative controls. Calibration bar is the same for all images (50 µm). (**c**) Quantitative representation of the data (*n* = 3/group). Asterisks (*) indicate a statistically significant difference (*p* < 0.05) compared with the b-SPM mock triton group. Black squares (■) indicate a statistically significant difference (*p* < 0.05) between the no triton and the triton groups.

**Figure 5 biomolecules-11-01187-f005:**
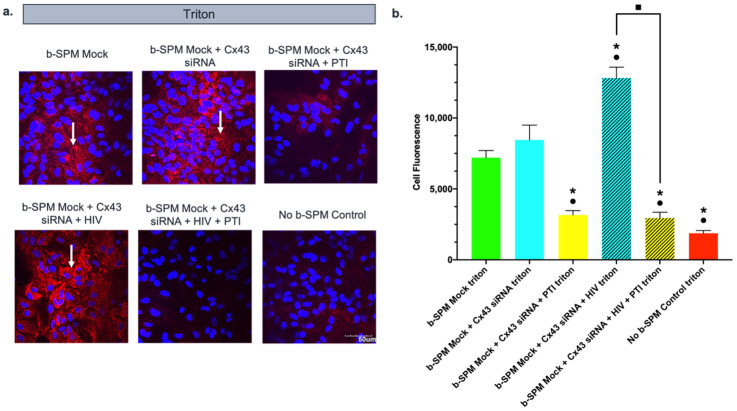
PTI significantly reduced the enhanced b-SPM uptake in b-SPM siRNA and HIV-Tat-treated astrocytes. Rho-avidin was used to visualize b-SPM (red), and DAPI was used to visualize the nuclei (blue). The presence of b-SPM presence is marked with the white arrows. (**a**) Qualitative representation of the data. Mock (only treated with HiPerFect) and mock + PTI astrocytes were incubated in either 80 µM of b-SPM or no b-SPM (PBS). No b-SPM samples were used as negative control. Calibration bar is the same for all images (50 µm). (**b**) Quantitative representation of the data (*n* = 3/group). Asterisks (*) indicate statistically significant differences (*p* < 0.05) with the b-SPM mock triton group. Black circles (•) indicate statistically significant differences (*p* < 0.05) with the b-SPM mock + Cx43 siRNA triton group. Black squares (■) indicate a statistically significant difference (*p* < 0.05) between the b-SPM mock + Cx43 siRNA + HIV triton and the b-SPM mock + Cx43 siRNA + HIV + PTI group.

**Figure 6 biomolecules-11-01187-f006:**
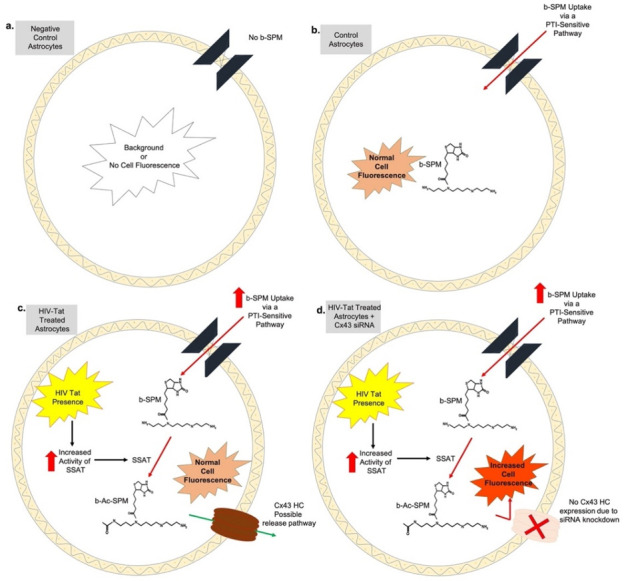
Summary model. (**a**) Negative control astrocytes were not exposed to b-SPM molecule (no b-SPM). The signal obtained from this group of negative control cells was considered to be the background or no cell fluorescence. (**b**) In control astrocytes, b-SPM is taken up by a PTI-sensitive uptake mechanism. (**c**) In HIV-Tat-treated astrocytes, there is increased activity of SSAT [27]. This leads to increased b-SPM uptake within the cell; b-SPM may be converted to Ac-SPM (perhaps b-Ac-SPM) and released. The overall levels of PAs within the cells are stable, suggesting that additional b-SPM is taken up into the cell to maintain a stable pool. (**d**) In the situation where astrocytes are treated with HIV-Tat and the Cx43 HCs are also knocked down using siRNA, the b-SPM accumulation as measured by increased cell fluorescence is more notable, suggesting that Cx43 HC are a release pathway for Pas, particularly in HIV-Tat-treated astrocytes. This increase in cell fluorescence may be due to the presence of b-SPM and/or “b-Ac-SPM” (via conversion of b-SPM to b-Ac-SPM due to SSAT being highly activated in HIV-Tat-treated astrocytes) and their inability to be released due to knockdown of Cx43 HC expression.

## Data Availability

The data supporting this manuscript is available upon request to the corresponding authors (Christian J. Malpica Nieves and Serguei Skatchkov).

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
