# Peer review of "Uptake of Biotinylated Spermine in Astrocytes: Effect of Cx43 siRNA, HIV-Tat Protein and Polyamine Transport Inhibitor on Polyamine Uptake"

_biomolecules, 2021, doi:10.3390/biom11081187_

Round 1
Reviewer 1 Report
Evaluation of the MS 1329497
Biomolecules
The paper of Malpica-Nieves et al. reports the use of biotinylated spermine (b-SPM) to analysed polyamines uptake in astrocytes after treatment with a siRNA against the hemichannel Cx43 (a polyamine transporter), the HIV-Tat protein (to mime HAND) and the novel polyamine transport inhibitor PTI (to block spermine flux). This experimental strategy is well fitting to shed light on the polyamine transport occurring between neurons and glia cells since, as recalled by the authors, in healthy brain polyamines are accumulated, but not synthesized, in astrocytes. Unpredictably, the main result obtained by this study is that polyamines uptake is via a transporter/channel other than Cx43-HCs, since astrocytes treated with both HIV-Tat and siRNA-Cx43 showed increased cellular accumulation of b-SPM. Furthermore, the utilization of PTI, which blocks the b-SPM uptake, was demonstrating that polyamine-uptake is via the PTI-sensitive transport mechanism defined as organic cation transporter (OCT). It was shown that Cx43 hemichannels (HCs) are not major pathway for b-SPM uptake in the condition of normal extracellular calcium concentration, but may be involved in the release of polyamines to the extracellular space during viral infection.
The work is very interesting and solid. The manuscript is well written. A great amount of experiments and data are reported. I do consider it an important contribution towards understanding how polyamine flux between neurons and astrocytes occurs in physiological conditions and in the pathological disorder HAND after HIV viral infection. The MS well adheres to the main points requested by Biomolecules policy.
There is nothing to edit, so I suggest accepting this manuscript as it is after addressing the potential roles for Ac-PAs in the etiopathogenesis of HAND in the section Discussion, which would underline the importance of the findings reported in the work. In my opinion, it has to be considered that the pathways for PAs uptake and release, and for Ac-PAs release, might also be relevant to design innovative pharmacological approaches to HAND. As a matter of fact, several molecules, including drugs in clinical use or under investigation have been reported to impact on the function of the Cx43 or the organic cation transporter OCT3 (for example see the reviews: Massmann et al, The organic cation transporter 3 (OCT3) as molecular target of psychotropic drugs: transport characteristics and acute regulation of cloned murine OCT3. Pflugers Arch - Eur J Physiol 2014, 466:517–527; Koepsell, Organic Cation Transporters in Health and Disease. Pharmacol Rev 2020, 72:253–319; Willebrords et al, Inhibitors of connexin and pannexin channels as potential therapeutics. Pharmacol Ther 2017, 180:144-160; Charveriat et al, Connexins as therapeutic targets in neurological and neuropsychiatric disorders. BBA Molecular Basis of Disease 2021, 1867:166098).
I encourage the authors to discuss and propose the OCT pathways as new targets to prevent/treat HAND, possibly also for drug repurposing.
I recommend publishing of this MS on Biomolecules with the minor revision proposed in Discussion and outlined above.
Author Response
Malpica et al., MDPI-Biomolecules, Special Issue, # 132497
Response to Reviewer 1:
We appreciate the positive evaluation of our study by Reviewer 1 who described the work as “very interesting and solid. The manuscript is well written. A great amount of experiments and data are reported…. There is nothing to edit, so I suggest accepting this manuscript as it is….”
As suggested, we have included new references and a paragraph discussing the possible therapeutic options to treat HAND based upon these findings and cross-purposing some of the compounds and drugs that are known to modulate OCT3 and Cx43.
The paragraphs added to the end of the discussion and in conclusion, they are the following:
In the Discussion:
The actions of the PAs, SPM and SPD, on neuronal receptors and channels has been well studied (20,21,29,32,37-40). Less is known about the actions of Ac-PAs and how increased release of Ac-PAs from astrocytes in HIV-infected individuals could contribute to HAND. Our study examining PA uptake and release in HIV-Tat treated astrocytes may lead to innovative pharmacological approaches to treat HAND. Blocking uptake of PAs will decrease the pool of PAs in astrocytes to convert to Ac-PAs, whereas blocking the release of Ac-PAs could prevent their actions on neuronal receptors and channels. There are currently many molecules, including drugs in clinical use or under investigation that have been reported to impact the function of Cx43 (we added new references: Goldberg et al., 1996 [Ref 67]; Willebrords et al., 2017 [Ref 68]; Charveriat et al., 2021 [Ref 69]) or the organic cation transporter OCT3 (Massmann et al., 2014 [Ref 70]; Koepsell, 2020 [Ref 71]). These drugs may be repurposed to alter the uptake of PAs into astrocytes and release of PAs and Ac-PAs from astrocytes with the potential to prevent/treat HAND.
In the Conclusion:
Interestingly, HIV-Tat induced Cx43 expression only in humans but not in rodent astrocytes [73] that is consistent with our data (not shown). Furthermore, treatment with HIV-Tat protein increases b-SPM uptake/accumulation when Cx43 function was blocked (siRNA treated astrocytes). There is also the possibility that PAs such as SPD and modified SPM (BEN-SPM, b-SPM) may activate antizyme-1 and SSAT [74] but if b-SPM does so, this will not affect (i) either b-SPM fluorescence accumulation or (ii) the effect of siRNA targeting Cx43. Finally, the OCT-3 transporter may be a major pathway for b-SPM uptake [70] since the SLC22A-3 is expressed in astrocytes [75] and is nonselective for many drugs, toxins, and endogenous compounds [40-43,70-71]. Therefore, our data suggest the possibility that Cx43 HCs are a release pathway for PAs in HIV-Tat transformed astrocytes since b-SPM accumulation is greater in the absence of Cx43 expression.
Thank you very much.

Reviewer 2 Report
In this paper, the authors investigated the mechanism of spermine uptake in astrocytes and tested the effect of Cx43 siRNA, HIV-Tat protein and polyamine transport inhibitor.
The spermine uptake activity was assessed using biotinylated spermine and fluorescent labeling. The substrate is not a natural one but can represent some aspects of spermine transport in astrocytes.
The authors found that Cx43 was not a major contributor for spermine uptake in astrocytes. The combination of siRNA for Cx43 and HIV-Tat protein treatments increased the accumulation of b-SPM. The authors concluded from the result to that HIV-Tat protein induced SSAT and acetylation of b-SPM, and Cx43 catalyzed the export of acetylated b-SPM from cells.
The paper is written clearly and the data presented support the author’s conclusion. There are some benefit for readers by providing the potential role of Cx43 and HIV-Tat protein in spermine transport.
The paper will be acceptable to biomolecules after minor modifications.
Minor points
1. It is likely that polyamine transport inhibitor induces antizyme during incubation of cells with inhibitor for 1h prior to the uptake assay, which blocks polyamine and potentially b-SPM uptake . Could authors add data or comments on the role of antizyme in the uptake of b-SPM?
2. The paper will be more convincing if the authors show acetylated polyamine or SSAT protein levels in HIV-Tat protein treated cells. A SSAT inducer such as BENSpm is a potential control to verify the model.
3. Does HIV-Tat protein affect the levels of Cx43 or PTI sensitive polyamine transporter?
4. Page1, line 41, there is no description for CNS.
Author Response
Malpica et al., MDPI-Biomolecules, Special Issue, # 132497
Response to Reviewer 2:
We thank the reviewer very much for the encouraging support of our research and specifically for the good suggestions/questions:
- It is likely that polyamine transport inhibitor induces antizyme during incubation of cells with inhibitor for 1h prior to the uptake assay, which blocks polyamine and potentially b-SPM uptake. Could authors add data or comments on the role of antizyme in the uptake of b-SPM?
We can only add comments since we do not have data on antizyme activity. Indeed, there is speculation in the literature about some possible cross-talk between uptake at the membrane and the synthesis/degradation of polyamines in the cytoplasm. Specifically, it was discussed that the synthesis of polyamines declines when cells obtain them by uptake. In addition, transporter(s) can be affected when polyamine synthesis is blocked (Gitto et al., 2018 [Ref 44] and Madan et al., 2016 [Ref 55]).
As we noted in our previous publication (Malpica et al., 2020 [Ref 33]), PA production in the cell is highly regulated and this regulation involves antizyme (AZ). When PA levels are high, these proteins can regulate themselves at the translational level, inhibiting ODC and exerting a negative control on PA uptake in the cell. AZ binds to ODC subunits and targets them for ubiquitin-independent degradation by the 26S proteasome. To regulate PA metabolism even further, the cells contain an antizyme inhibitor (AZI) which is an ODC-related protein and helps to negate AZ function (Kahana 2009; Reddy 2015; Ramos-Molina et al. 2018).
In the present study, when we treat with PTI the PA levels inside the cell may be reduced, therefore, it is more likely that the enzyme that will be induced is AZI since there is no source of PAs from the uptake pathway. The cell will only rely on the synthesis of PAs by inhibiting AZ and allowing PA synthesis. So in theory the role of AZ is to be inhibited by AZI to promote PA synthesis.
Based on the data in the current manuscript, we do not know yet if synthesis and antizyme or antizyme inhibitor activity is changing, but this is not a goal of the current study.
- The paper will be more convincing if the authors show acetylated polyamine or SSAT protein levels in HIV-Tat protein treated cells. A SSAT inducer such as BENSpm is a potential control to verify the model.
Thank you very much for the recommendation, we added new references and this is something we plan to do in the future, however, this is out of the scope of the current manuscript. The recommendation to use BENSpm in these future experiments is a good one.
- Does HIV-Tat protein affect the levels of Cx43 or PTI sensitive polyamine transporter?
- a) HIV-Tat effect on Cx43. We previously performed experiments to confirm the findings of Berman et al., 2016 [Ref 73] who described that HIV-Tat increases Cx43 expression in human astrocytes, but not in rodent astrocytes. Consistent with their statement, HIV-Tat treatment did not increase Cx43 expression in control mouse astrocytes (please, see the figure attached in the document).
We included the following sentences in the conclusion section:
Interestingly, HIV-Tat induced Cx43 expression only in humans but not in rodent astrocytes [73] that is consistent with our data (not shown). Furthermore, treatment with HIV-Tat protein increases the b-SPM uptake/accumulation when Cx43 function was blocked (siRNA treated astrocytes).
- b) HIV-Tat effect on PTI sensitive polyamine transporter.
The PTI sensitive transporter(s) have not yet been identified. Until we identify the specific transporter which may be the OCT3 (SLC22A-3) transporter, it will be difficult to determine the effect of HIV-Tat on the transporter. Some membrane transporters (Inazu et al., 2003 [Ref 43]) such as SLC22A-1,-2,-3 ( Sala-Rabanal et al., 2013 [Ref42]; Li et al., 2015 [Ref 40]; Cui et al., 2009 [Ref 58]) are found in neurons and astrocytes but they are polyspecific and carry all types of poly-and mono-amines (Sala-Rabanal et al., 2013 [Ref 42]).
- Page1, line 41, there is no description for CNS.
We corrected this and it now reads “……HIV strains isolated from HIV infected central nervous system (CNS).”
Thank you very much.
